

# Pollution and health risk assessment of rare earth elements in *Citrus sinensis* growing soil in mining area of southern China

Jinhu Lai[1], Jinfu Liu[2], Daishe Wu[1,3] and Jinying Xu[1]

[1] School of Resources and Environment and Key Laboratory of Poyang Lake Environment and Resource Utilization, Ministry of Education, Nanchang University, Nanchang, China
[2] Nanchang Institute of Technology, Nanchang, China
[3] Pingxiang University, Pingxiang, China

## ABSTRACT

**Background**. Analyzing the pollution and health risk of rare earth elements (REEs) in crop-growing soils around rare earth deposits can facilitate the improvement of REE mining-influenced area. In this study, pollution status, fraction and anomaly, plant accumulation characteristics, and potential risks of REEs (including heavy and light rare earth elements, HREEs and LREEs) in *C. sinensis* planting soil near ion-adsorption deposits in southern Ganzhou were analyzed. The influence of the soil environment on REEs in soil and fruit of *C. sinensis* was also explored.

**Methods**. The geo-accumulation index ($I_{geo}$) and ecological risk index(RI) were used to analyze the pollution potential and ecological risks of REEs in soils, respectively. Health risk index and translocation factor (TF) were applied to analyze the accumulation and health risks of REEs in fruit of *C. sinensis*. The influence of soil factors on REEs in soil and fruit of *C. sinensis* were determined *via* correlation and redundancy analysis.

**Results**. Comparison with background values and assessment of $I_{geo}$ and RI indicated that the soil was polluted by REEs, albeit at varying degrees. Fractionation between LREEs and HREEs occurred, along with significant positive Ce anomaly and negative Eu anomaly. With TF values $< 1$, our results suggest that *C. sinensis* has a weak ability to accumulate REEs in its fruit. The concentrations of REEs in fruit differed between LREEs and HREEs, with content of HREE in fruit ordered as Jiading > Anxi > Wuyang and of LREE in fruit higher in Wuyang. Correlation and redundancy analysis indicated that $K_2O$, $Fe_2O_3$ and TOC are important soil factors influencing REE accumulation by *C. sinensis*, with $K_2O$ positively related and $Fe_2O_3$ and TOC negatively related to the accumulation process.

# INTRODUCTION

Rare earth elements (REEs), containing lanthanide elements (lanthaum (La), cerium (Ce), praseodymium (Pr), neodymium (Nd), samarium (Sm), europium (Eu), gadolinium (Gd), terbium (Tb), dysprosium (Dy), holmium (Ho), erbium (Er), thulium (Tm), ytterbium

Corresponding authors
Daishe Wu, dswncu@126.com
Jinying Xu, xujy2020@ncu.edu.cn

(Yb), lutetium (Lu)), scandium (Sc) and yttrium (Y), have similar physicochemical properties and generally co-exist in the natural environment (*Lian et al., 2022*; *Pereira et al., 2022*). Based on their atomic numbers and masses, REE can be further classified as light rare earth elements (LREEs, including La, Ce, Pr, Nd, Sm, and Eu) and heavy rare earth elements (HREEs, including Gd, Tb, Dy, Ho, Er, Tm, Yb, and Lu) (*Lian et al., 2022*). Generally, REEs are widely used in high-tech fields such as military equipment, artificial intelligence, metallurgical machinery, and medicine (*Yuan et al., 2017*; *Liu et al., 2021*). The global demand for REEs, especially HREEs, is increasing rapidly (*Huang et al., 2021*; *Liu et al., 2021*). This leads to an increasing exploitation of REE minerals, especially the ion-adsorption REE deposits, which are the main sources of HREEs. The mining of such deposits, using methods of pool leaching, heap leaching, or *in-situ* leaching, causes the destruction of environmental components (*Li, Xu & Li, 2020*), which results in the spreading of REEs to the surrounding agricultural soil. Because of their characteristics similar to those of heavy metals, excess REEs accumulated in soils can be transported along the food chain, resulting in risks to environmental and human health (*Pereira et al., 2022*; *Lian et al., 2022*). Therefore, it is of great necessity to study the pollution, transfer, and potential risk of REEs in agricultural soils around mining areas.

Agricultural plants play a vital role in the transfer of metals from soil to humans (*Kotelnikova et al., 2020*), and various studies have analyzed the accumulation and potential health risks of REEs in agricultural plants (*Arbalestrie et al., 2022*; *Liu et al., 2022*; *Wiche & Heilmeier, 2016*; *Zhuang et al., 2017*). *Wiche & Heilmeier (2016)* investigated the REE accumulation ability of different plants and reported that herbaceous species, such as *Brassica napus*, can accumulate significantly higher La and Nd concentrations compared with grasses (*Zea mays*, *Hordeum vulgare*, among others), whereas the Ge concentrations were higher in grasses than in herbs. *Zhuang et al. (2017)* indicated that vegetable crops planted in agricultural soils around mining areas accumulated significantly higher amounts of REEs compared with control crops, although the associated health risk was low. However, the accumulation potential and health risks of REEs in plants depend on the soil environment (*Lin et al., 2022*; *Moreira et al., 2019*; *Wiche & Heilmeier, 2016*). For example, acidic soil can significantly increase the uptake of REEs by plants (*Wiche & Heilmeier, 2016*), and soil containing large amounts of organic matter in the form of low-molecular organic acids may facilitate the translocation of REEs to plants (*Liu et al., 2021*). It is therefore crucial to clarify the accumulation potential and health risks of REEs in agricultural crops in different soils affected by REE deposits.

Southern Ganzhou, located in the southern of Jiangxi Province in China, is the main producing area of rare earths, which reserves 1/3 of the HREEs in China and can provide 70% of ion-adsorption REEs that China produces and sales (*Yuan et al., 2019*; *Li, Xu & Li, 2020*). However, with the over exploitation of the deposits during the last decades, the surrounding agricultural soil are threaten by REEs (*Li, Xu & Li, 2020*; *Huang et al., 2021*; *Shi et al., 2022*). Southern Ganzhou is also the hometown of *Citrus sinensis*, which has the *C. sinensis* planting area reaching 163 million acres. Studies have indicated that *C. sinensis* can accumulate a certain amount of REEs in the pulp (*Cai & Rui, 2013*; *Cheng et al., 2015*). However, investigations on the pollution risk and accumulation potential of REEs in the

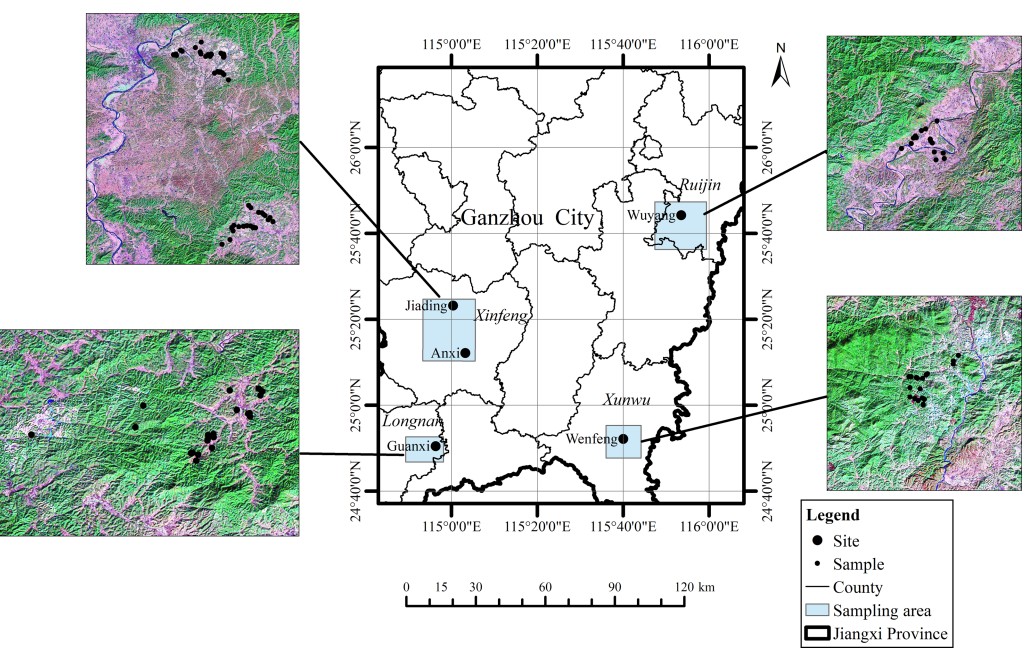

**Figure 1** The study area and sampling sites of soils and plants.

soil; the *C. sinensis* system of southern Ganzhou are rare. In this study, different *C. sinensis* planting areas around the REE deposits of southern Ganzhou were investigated, with the following objectives: (1) to determine the REE pollution status of *C. sinensis* planted soil; (2) to indicate the REE accumulation and health risks in fruits of *C. sinensis*; (3) to elucidate the influence of the soil environment on REEs in soil and pulp of *C. sinensis*. The results of this study can provide a scientific reference for the management of REE deposits affecting agricultural fields.

# MATERIAL AND METHODS

## Study area

Southern Ganzhou (24°29′N–27°09′N, 113°54′–116°38′) is located in the south of Jiangxi Province, China (Fig. 1). Because of its climatic and soil conditions, this region has become the most important *C. sinensis* cultivation area in China. Southern Ganzhou also contains the highest amounts of ion-adsorption rare earth deposits in China and is therefore known as "the Kingdom of Rare Earths". The REEs is mainly distributed in the cities of Longnan, Xunwu, Xinfeng, Ningdu, Dingnan, Anyuan and Quannan. Longnan city is dominated by heavy rare earth rich in Y, Xunwu city is dominated by light rare earth lacking Y and the other cities are dominated by rare earth with abundant Eu and medium Y. The exploitation of these deposits has led to higher concentrations of REEs in the soil of southern Ganzhou compared to the nationwide average values.

## Sample collection and analysis

For this study, the cities of Xinfeng, Longnan, and Xunwu were selected for sampling as they contain different types of rare earths deposits (Fig. 1). The City of Ruijin, which is not the main distribution area of rare earths in southern Ganzhou, was used as control. Two sampling sites were established in Xinfeng City, one in Jiading Town and one in Anxi Town. One sampling site was established in Longnan, Guixi Town, one in Xunwu, Wenfeng Town, and one in Ruijin, Wuyang Town. Surface soil under *C. sinensis* was sampled at all sites, using a stain-less shovel. To determine the accumulation potential and health risks of REEs in the fruit of *C. sinensis* in sites with different REE pollution characteristics, sampling sites (Jiading Town, Anxi Town, and Wuyang Town) located in Xinfeng City and Ruijin City were chosen for *C. sinensis* fruit sampling. At each sampling site, 15 replicated samples were collected and stored in polyethylene bags for for further analysis.

The soil samples and the pulp of the fruits were air-dried, ground and sieved for subsequent analysis. The LREEs (La, Ce, Pr, Nd, Sm, Eu) and HREEs (Gd, Tb, Dy, Ho, Er, Tm, Yb, Lu) in soil and fruit were analyzed *via* inductively coupled plasma-mass spectrometry (ICP-MS, iCAP Q, ThermoFisher, USA). To ensure quality of data, blank control, duplicate samples, and standard reference materials were used. The blank control produced values less than 3% of the measured values. Triplicate analysis was conducted for determining the concentrations of the REEs to ensure reliable results. The REE concentration was the mean concentration of the triplicate samples. The procedure of analysis was checked with a certified standard of GBW07358 (Chinese national geo-standard), which is widely used for quality assurance in geochemical analysis.

The soil pH was measured by using a pH meter in a 1:2.5 soil: water mixture, soil total organic carbon (TOC) was determined by a Torch TOC combustion analyzer. The metal oxides ($Fe_2O_3$, $CaO$, $Na_2O$, $K_2O$) were determined *via* X-ray Fluorescence Spectrometry (EDX-8000; XRF, Shimadzu, Japan).

## Pollution and ecological risk assessment of REE in soil
### Pollution assessment
The pollution status of the studied soil was assessed by the geo-accumulation index ($I_{geo}$), using Eq. (1):

$$I_{geo} = \log 2 \left[ \frac{C_i}{1.5B_i} \right] \tag{1}$$

where, $C_i$ is the concentration of measured REE $i$; $B_i$ is the background concentration of REE $i$, which refers to the background values of REE in the soil of Jiangxi Province, China (*CNEMC, 1990*). The $I_{geo}$ can be classified into five categories as shown in Table 1.

### Potential ecological risks of REEs in soil
The potential ecological risk (RI) of REEs in soil was assessed with the methods provided by *Hakanson (1980)*. This method can be used to identify the ecological risks of both single element and combined elements, using Eqs. (2), (3) and (4):

$$C_f^i = C_i / C_b^i \tag{2}$$

**Table 1 The classes of Geo-accumulation index ($I_{geo}$) and ecological risk index (RI).**

| $I_{geo}$ Class[a] | Soil quality | $E_r^i$ class | Potential risk | RI class[b] | Ecological risk |
|---|---|---|---|---|---|
| <0 | Unpolluted | $E_r^i < 40$ | Low | RI < 150 | Low |
| 0–1 | Unpolluted to moderately polluted | $40 \leq E_r^i < 80$ | Moderate | $150 \leq$ RI $< 300$ | Moderate |
| 1–2 | Moderately polluted | $80 \leq E_r^i < 160$ | Considerable | $300 \leq$ RI $< 600$ | Considerable |
| 2–3 | Moderately to highly polluted | $160 \leq E_r^i < 320$ | High | RI $\geq 600$ | Very high |
| 3–4 | Highly polluted | $E_r^i \geq 320$ | Very high | | |
| 4–5 | Highly to very highly polluted | | | | |
| >5 | Very highly polluted | | | | |

$$E_r^i = C_f^i \cdot T_r^i \tag{3}$$

$$RI = \sum E_r^i \tag{4}$$

where, $C_f^i$ and $E_i$ refers to the contamination factor and ecological risk factor, respectively of element $i$ in soil, respectively; $C_i$ is the concentration of element $i$ in soil; $C_b^i$ and $T_r^i$ refers to the pre-industrial background value and toxic response factor of element $i$ in soil, respectively. The classes of $E_r^i$ and RI are shown in Table 1.

### Anomaly calculation of REEs in soil

Geochemical parameters can exactly reflect the enrichment, fraction and sources of REE in soils (*Lian et al., 2022*). In this study, the Eu anomalies ($\delta Eu$) and Ce anomalies ($\delta Ce$) were used to represent the geochemical parameters of REE in soils, which can be calculated as Eqs. (5) and (6):

$$\delta Eu = Eu_N / (Sm_N \times Gd_N)^{1/2} \tag{5}$$

$$\delta Ce = Ce_N / (La_N \times Pr_N)^{1/2}. \tag{6}$$

The North America Shale Composite (NASC) was employed to normalize the concentrations of REEs to reduce the zigzag patterns of REEs in the soil. Here, the $Eu_N$, $Sm_N$, $Gd_N$, $\delta Ce$, $La_N$ and $Pr_N$ represents the NASC normalized values of measured concentrations of measured Eu, Sm, Gd, Ce, La, and Pr in soil, respectively. Values of $\delta Eu$ and $\delta Ce$ higher than 1 indicate a positive anomaly of Eu and Ce, and values lower than 1 indicate a negative anomaly of Eu and Ce.

The fractionation of LREE and HREE was characterized by LREE/HREE, the NASC normalized La/Sm $((La/Sm)_N)$ and Gd/Yb $((Gd/Yb)_N)$. The $(La/Sm)_N$ and $(Gd/Yb)_N$ can be calculated as follows:

$$(La/Sm)_N = \frac{La_{sample}/La_N}{Sm_{sample}/Sm_N} \tag{7}$$

$$(Gd/Yb)_N = \frac{Gd_{sample}/Gd_N}{Yb_{sample}/Yb_N} \tag{8}$$

where, $La_{sample}$, $Sm_{sample}$, $Gd_{sample}$ and $Yb_{sample}$ refers to the concentration of La, Sm, Gd and Yb in the studied soils, respectively. $La_N$, $Sm_N$, $Gd_N$ and $Yb_N$ refers to the NASC normalized values of La, Sm, Gd, and Yb, respectively.

## Accumulation and health risks of REEs in fruit

### Accumulation process

The accumulation process of REEs from soil to fruit was assessed using the translocation factor (TF), as described in Eq. (9):

$$TF = C_{fruit}/C_{soil}. \tag{9}$$

where, $C_{soil}$ and $C_{fruit}$ represents the concentrations of REEs in soil and fruit, respectively. A TF > 1 indicates a high accumulation of REEs in plant.

### Health risk assessment

The human health risk of REEs in fruits of *C. sinensis* was assessed using the health risk model provided by the US Environmental Protection Agency (*Lian et al., 2022*):

$$ADD_i = \frac{C_i \times GW \times EF \times ED}{BW \times AT} \tag{10}$$

where $ADD_i$ is the average daily dose of element i (mg/kg/day); BW is the body weight (60 kg); AT refers to the average time (365 ×70 days); $C_i$ is the mean concentration of element i in the soil (mg/kg); GW refers to the daily intake amount of fruit (kg/day), which was identified as 0.269 kg/day according to the *Piao et al. (2021)*; EF refers to exposure frequency (180 days/year); ED refers to the exposure duration (70 years).

## Statistical analysis

One-way analysis of variance (ANOVA) was applied to identify the difference of REEs at different sampling sites in the study area. Prior to ANOVA, a Shapiro–Wilk test and Leven's test were conducted to ensure the normality and homogeneity of the data. Difference was significant at $p < 0.050$ or $p < 0.010$. Correlation and redundancy analysis were employed to explain the influence of soil factors on REEs in soils and fruit. All statistical analyses were conducted using SPSS 26.0, and the figures were plotted by R v3.5.1.

# RESULTS AND DISCUSSION

To clarify the impacts of REEs in *C. sinensis* fruit and soil around the rare earth deposits in southern Ganzhou on human and environmental health, we analyzed the geochemical characteristics, pollution status, and ecological risks of REEs in soils, determined the accumulation and health risks of REEs in *C. sinensis* fruit, and explored the main soil factors influencing REE accumulation in soils and *C. sinensis* fruit.

## The basic information of soil factors

The soil factors differed significantly among the sampling sites (Table 2, $p < 0.010$). Generally, pH was highest in Anxi and Wenfeng, with the average values of 4.82 and 4.57, respectively. Whereas, Guanxi and Wenfeng were characterized by significantly higher TOC concentrations. The concentration of $Na_2O$ was significantly higher in Wenfeng, with the mean value of 0.612%. $K_2O$ had significantly higher values in Jiading, with the average value of 4.07%. The content of $Fe_2O_3$ was significantly higher in Anxi and Guanxi (mean values: 6.62% and 5.77%, respectively).

**Table 2  The basic information of soil factors in studied soils.**

| Site | pH | TOC (%) | Fe$_2$O$_3$ (%) | Na$_2$O (%) | K$_2$O (%) |
|------|-----|---------|----------------|-------------|------------|
| Jiading | 5.11[b] ± 0.562 | 0.793[a] ± 0.489 | 4.15[a] ± 0.28 | 0.165[a] ± 0.063 | 4.07d ± 1.31 |
| Anxi | 4.82[a] ± 0.370 | 1.14[b] ± 0.391 | 6.62[b] ± 1.42 | 0.095[a] ± 0.029 | 1.3[a] ± 0.655 |
| Guanxi | 5.02[b] ± 0.502 | 1.58[c] ± 0.447 | 5.77[b] ± 4.40 | 0.176[a] ± 0.087 | 2.67[c] ± 1.25 |
| Wenfeng | 4.57[a] ± 0.231 | 1.48[c] ± 0.319 | 2.58[a] ± 0.69 | 0.612[b] ± 0.536 | 1.74[ab] ± 1.50 |
| Wuyang | 5.8[c] ± 0.880 | 1.16[b] ± 0.401 | 3.75[a] ± 1.08 | 0.241[a] ± 0.187 | 2.41[bc] ± 0.585 |

**Notes.**
Lowercase letters (a, b, c) indicate the significance of soil factors' differences among sites.

## Spatial distribution of REEs in soil

The mean concentration of REE in soil of the study area is shown in Table 3. The mean concentration of La, Ce, Pr, Nd, Sm, Eu, Gd, Tb, Dy, Ho, Er, Tm, Yb and Lu was 64.3, 114, 14.2, 51.1, 9.82, 1.59, 9.15, 1.47, 8.88, 1.75, 5.05, 0.798, 5.15 and 0.817 ug/g, respectively, which were apparently higher than the national average values of REE in soils of China and the background values of REEs in soil of Jiangxi Province, China. Additionally, the mean concentrations of these REEs were higher than those of the world average values in soils, except for Pr, Eu, Tb, Ho, Tm and Lu. This indicates that the studied soil was polluted by the REEs to some extent, which is in agreement with previous studies on REEs in the soils of southern Ganzhou (*Yuan et al., 2019*; *Zhang et al., 2022*). The pollution of the *C. sinensis* planting soil detected in this study coincides with the large-scale exploitation of the surrounding rare earth deposits during the last decades (*Li, Xu & Li, 2020*). Besides, heap and pool leaching method, which is frequently used in southern Ganzhou and produce quantitative wastewater and tailing sand containing REEs, further aggravating soil REE pollution (*Wen, 2012*; *Xu, Li & Li, 2021*).

The spatial distribution of REEs in the study area is shown in Fig. 2. The concentrations of REEs were significantly different among sampling sites (Fig. 2, $p < 0.01$). Overall, the average concentrations of all REEs were significantly higher in Jiading and Guanxi. According to *Li (2012)*, Xinfeng County has the largest amount of middle yttrium and rich Eu style REE mining area in southern China, which may lead to high concentrations of REE in in Jiading. In addition, the exploitation time is closely related to the concentration of REEs in soil, and the shorter the exploitation time, the higher the migration capacity of REEs with water (*Wen, 2012*). Therefore, the shorter exploitation time of REE minerals in Xinfeng County may also contribute to a higher amount of REEs in Jiading (*Wen, 2012*). Similarly, Longnan County owns over 70% of the rich yttrium style REE mining area of the world, and the high-intensity exploitation of this minerals may lead to a higher amount of REEs in Guanxi (*Securities Daily, 2008*). Additionally, although both located in Xinfeng County, Anxi had significantly lower REE concentrations compared to Jiading (Fig. 2), most likely because of the lower intensity of rare earth exploitation in this region.

## Geochemical parameters of REEs in studied soils

The fractionation of REEs in *C. sinensis* planting soil was characterized by the LREE/HREE, (La/Sm)$_N$, (Gd/Yb)$_N$ (Table 4). The LREE/HREE values ranged from 4.89 to 12.1 in

**Table 3** The mean concentrations of REE in the study area and the comparison with other soils.

| REE | Mean values in this study (ug/g) | National average values in soils (ug/g) | The background values in soil of Jiangxi Province (ug/g) | World average values in soils (ug/g) |
|---|---|---|---|---|
| La | 64.3 | 37.5 | 45.0 | 50.0 |
| Ce | 114 | 64.7 | 79.9 | 7.00 |
| Pr | 14.2 | 6.67 | 10.3 | 35.0 |
| Nd | 51.1 | 25.1 | 33.3 | 4.50 |
| Sm | 9.82 | 4.94 | 6.64 | 1.00 |
| Eu | 1.59 | 0.980 | 1.02 | 4.00 |
| Gd | 9.15 | 4.38 | 6.01 | 0.700 |
| Tb | 1.47 | 0.580 | 0.900 | 5.00 |
| Dy | 8.88 | 3.93 | 6.27 | 0.600 |
| Ho | 1.75 | 0.830 | 1.22 | 2.00 |
| Er | 5.05 | 2.42 | 3.90 | 0.600 |
| Tm | 0.798 | 0.350 | 0.49 | 3.00 |
| Yb | 5.15 | 2.32 | 3.40 | 0.400 |
| Lu | 0.817 | 0.350 | 0.510 | 40.0 |

**Table 4** The geochemical parameters of rare earth elements in studied soils.

| | $\delta Ce$ | $\delta Eu$ | LREE/HREE | $(La/Sm)_N$ | $(Gd/Yb)_N$ |
|---|---|---|---|---|---|
| Jiading | 0.713 ± 0.326 | 0.640 ± 0.011 | 7.73 ± 1.86 | 4.49 ± 0.446 | 1.48 ± 0.205 |
| Anxi | 0.995 ± 0.190 | 0.757 ± 0.065 | 11.3 ± 3.62 | 5.06 ± 1.067 | 1.72 ± 0.425 |
| Guanxi | 1.11 ± 0.681 | 0.351 ± 0.191 | 6.52 ± 1.31 | 3.54 ± 0.771 | 1.28 ± 0.337 |
| Wenfeng | 2.80 ± 2.760 | 0.324 ± 0.141 | 11.1 ± 3.83 | 3.89 ± 0.919 | 1.46 ± 0.434 |
| Wuyang | 1.07 ± 0.145 | 0.594 ± 0.045 | 8.66 ± 1.60 | 4.35 ± 0.534 | 1.41 ± 0.334 |

Jiading (mean value: 7.73), 6.17 to 17.6 in Anxi (mean value: 11.3), 4.43 to 8.34 in Guanxi (mean value: 6.52), 5.15 to 21.2 in Wenfeng (mean value: 11.08), 5.17 to 11.5 in Wuyang (mean value: 8.66). This suggests that fractionation between LREE and HREE occurred in all sampling sites, with Anxi showing the largest degree of fractionation. All sampling sites had $(La/Sm)_N$ values significantly higher than those of $(Gd/Yb)_N$, indicating that the single fractionation degreee of LREEs was higher than that of HREEs. Fractionation of LREE and HREE may have influenced the geochemical characteristics of the studied soils (*Lian et al., 2022*).

The anomaly of REEs was characterized by $\delta Ce$ and $\delta Eu$ (Table 4). Based on the results, the $\delta Ce$ values were higher than 1 in Guanxi, Wenfeng and Wuyang. With trivalent and tetravalent valences in soil, Ce tends to show a behavoir different from that of the other REEs (*Lian et al., 2022*; *Liu et al., 2020*). It can be oxidized from Ce(III) to Ce(IV) in an oxidizing environment, thus becoming insoluble in soil in the form of $CeO_2$ *via* adsorption on Fe/Mn oxides (*Bispo et al., 2021*; *Manoj, Thakur & Prasad, 2016*). According to a previous study,
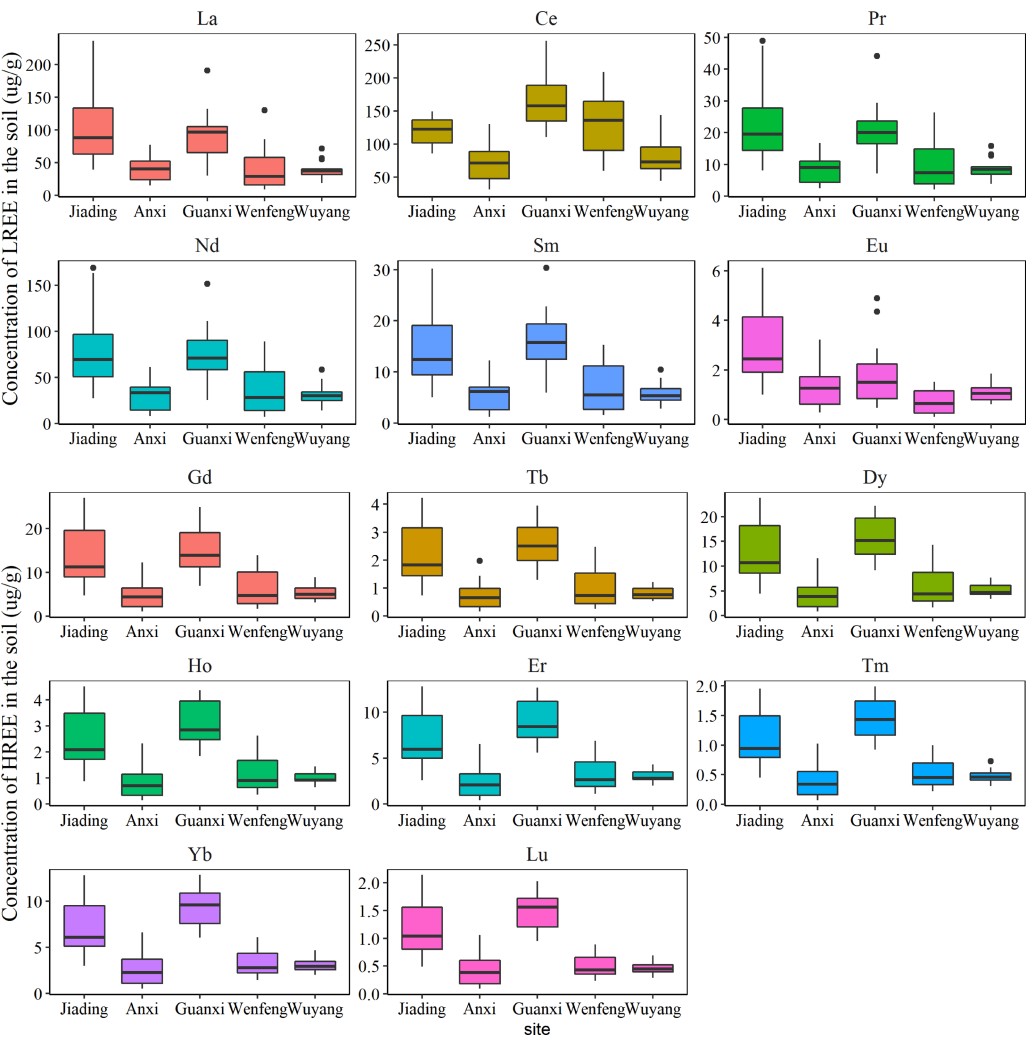

**Figure 2** The spatial distribution of rare earth elements in soils (HREE, heavy rare earth element; LREE, light rare earth element).

a different oxidation environment can be formed during red soil formation (*Yang et al., 1999*), which may lead to the sedimentation of Ce and the loss of other REEs in the soil, as observed for Guanxi, Wenfeng, and Wuyang. In contrast, the δEu values were below 1 at all sampling sites, with mean values of 0.640, 0.757, 0.351, 0.324, and 0.594 in Jiading, Anxi, Guanxi, Wenfeng, and Wuyang, respectively (Table 4), indicating a negative anomaly of Eu in the studied soils. The oxidation of insoluble Eu(II) to dissolved Eu(III) in different oxidation environments may have led to the Eu fractionation (*Lian et al., 2022*; *Yang et al., 1999*).

## Pollution and ecological risks of rare earth elements in soils

The pollution status of REEs was assessed using $I_{geo}$ (Fig. 3). The $I_{geo}$ values of REEs were significantly higher in Jiading and Guanxi (Fig. 3), with average values at these two sites higher than 0, indicating that Jiading and Guanxi were unpolluted to moderately polluted

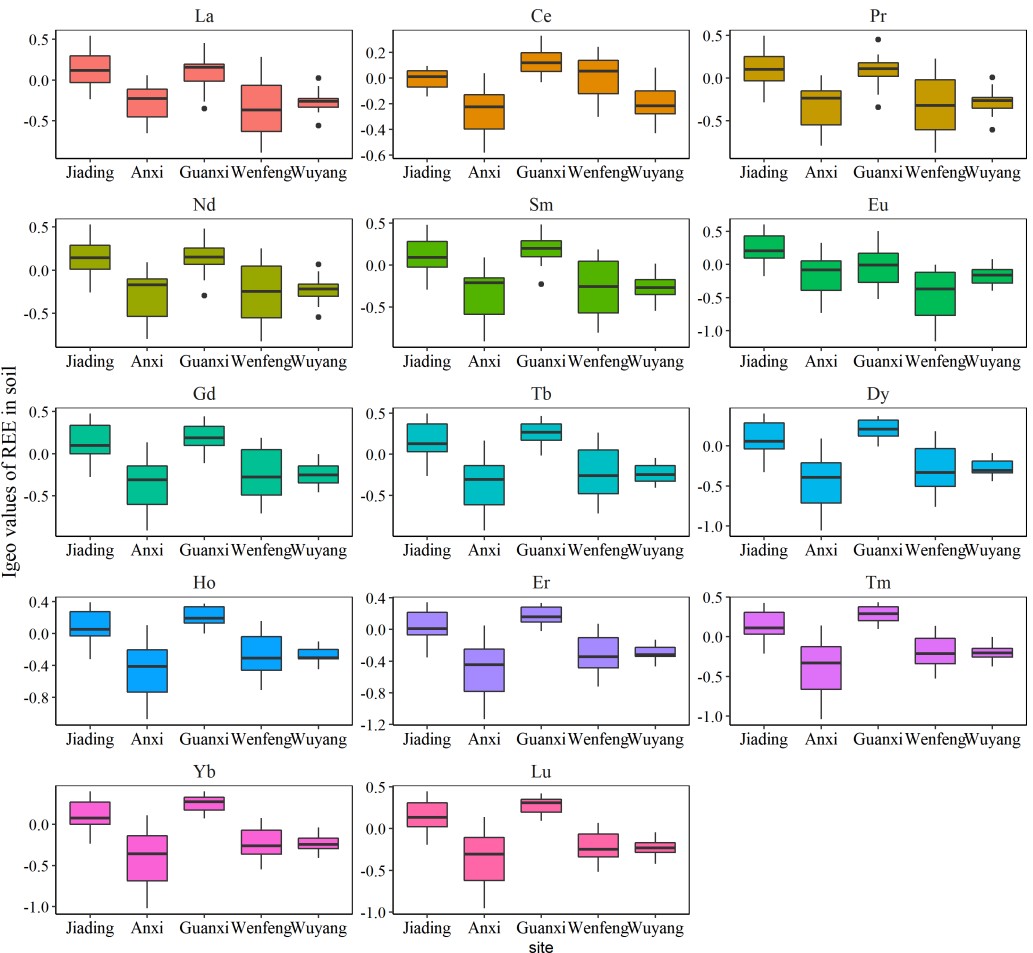

**Figure 3** **The geo-accumulation index (Igeo) of rare earth elements in studied soils.**

by REEs. This coincides with the spatial distinctions of the concentrations of REE (Fig. 2). Generally, metals from the natural sources do not cause environmental pollution, and the pollution detected in Jiading and Guanxi further highlight the influence of rare earth mining activities on the surrounding agricultural soil as discussed above.

The ecological risk of REEs is shown in Fig. 4. Based on the results, all REEs had Er values lower than 40.0, with the exception of Lu, which had Er values higher than 40.0 in Jiading and Guanxi (average Er value: 47.0 and 54.5, respectively), indicating a moderate ecological risk of Lu at these two sites. Additionally, REEs in Jiading and Guanxi had RI values above 150 (219 in Jiading, 241 in Guanxi), indicating a moderate ecological risk of REEs in these regions. This coincides with the significantly higher concentrations and pollution status of REE at these two sites as discussed above. In summary, the $I_{geo}$ and RI assessment showed that the soil planting *C. sinensis* in Jiading and Guanxi should pay special attention to.

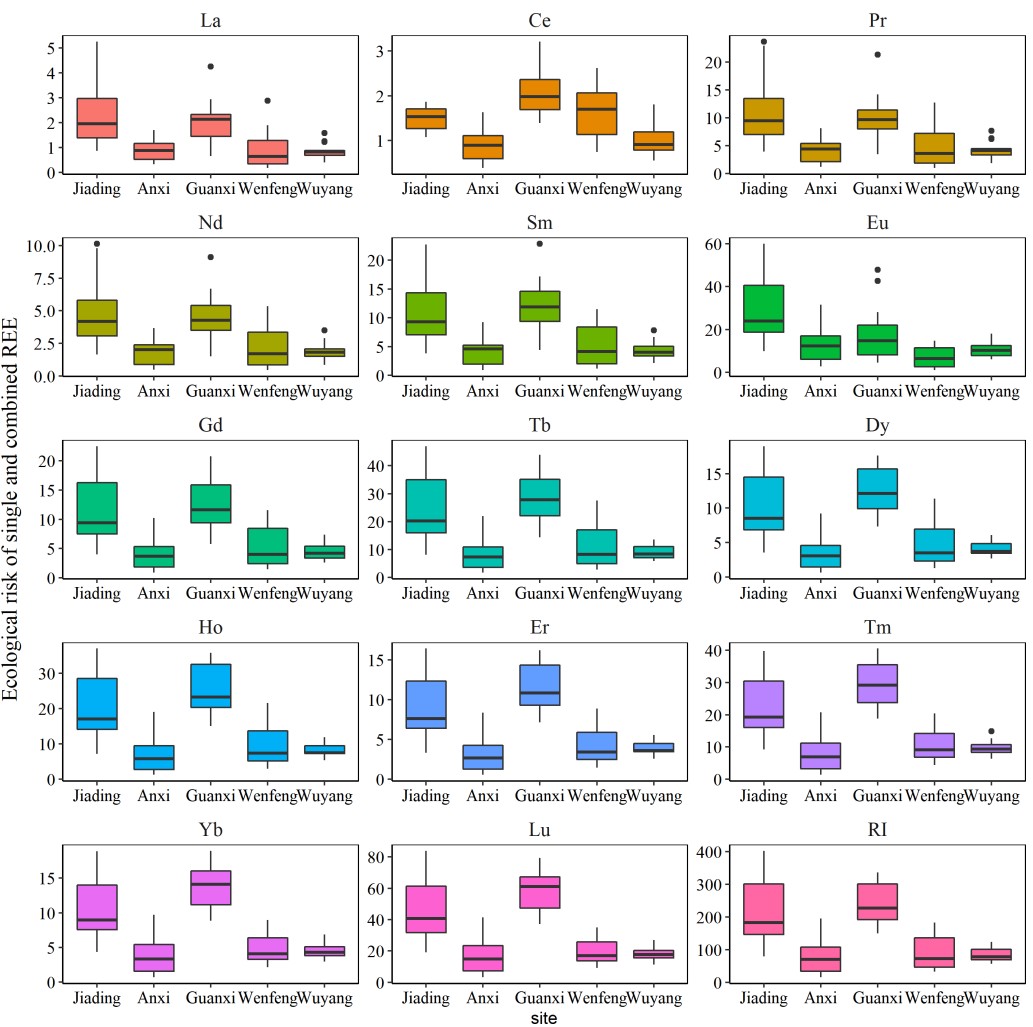

**Figure 4** **The potential ecological risk of rare earth elements in studied soils.**

## Accumulation and health risk of rare earth elements in *C. sinensis*

The concentrations of REEs in the fruit of *C. sinensis* are shown in Fig. 5. The concentrations of HREEs in fruit followed the order Jiading > Anxi > Wuyang, which agrees with the significant higher concentrations of HREEs in soil of Jiading (Fig. 2). A previous study also indicated a positive relationship between metals in plants and soils (*Ran et al., 2016*). The concentrations of LREES in fruits were generally higher in Wuyang ($p < 0.01$), which is not in line with the significantly lower REE concentrations in the soil from Wuyang (Fig. 2). Previous studies have suggested that it is not the total amount but the fraction of metals that detemines their accumulation in plants (*Xu et al., 2020*; *Xu et al., 2021a*). In Wuyang, the specific soil conditions may increase the solubility of LREEs, facilitating their accumulation of in fruits. The following part will further analyze the influence of soil factors on the REE accumulation of *C. sinensis*.

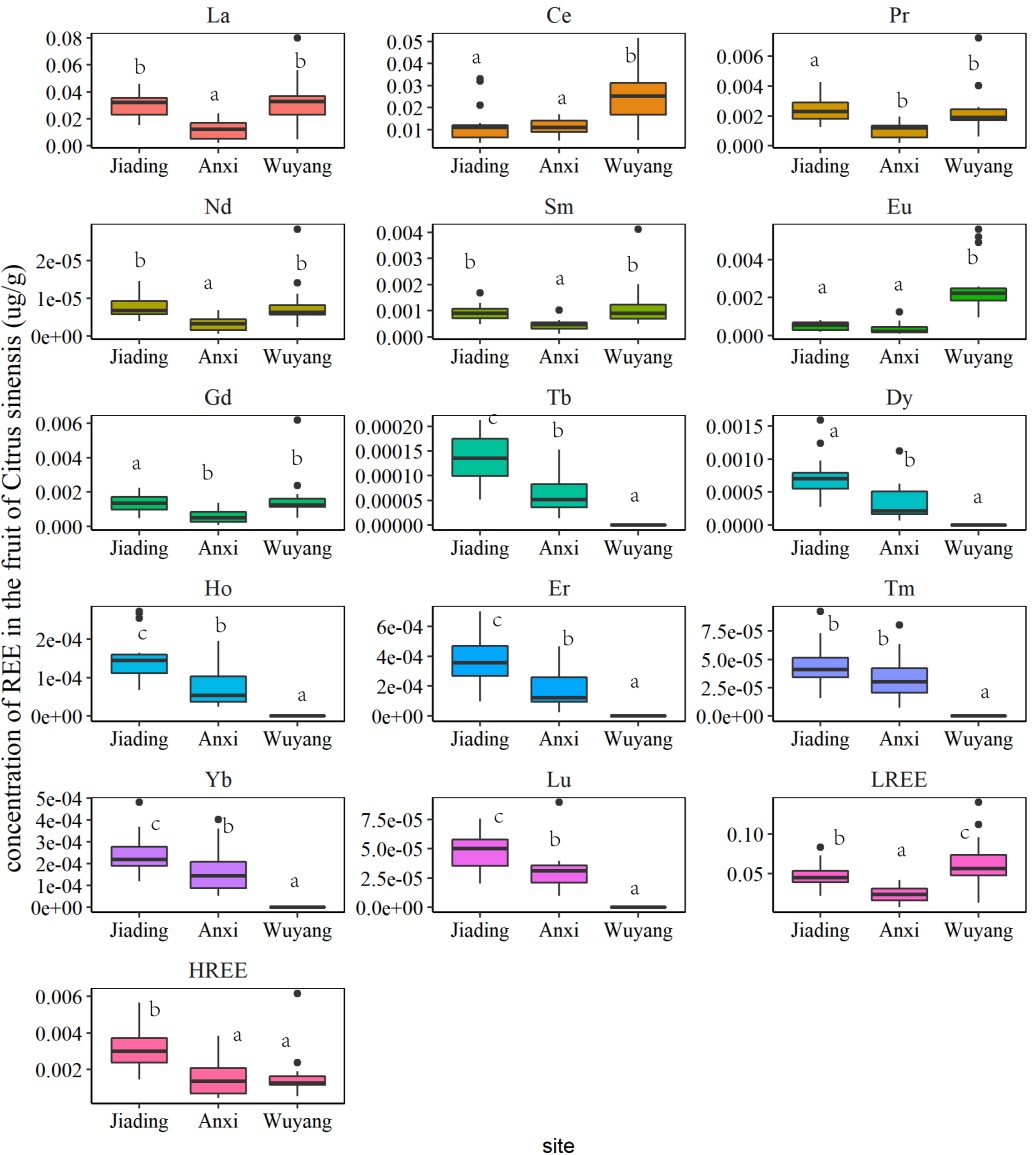

**Figure 5** **The concentrations of rare earth elements in the fruit of *C. sinensis*.** Different lowercase letters (a, b, c) indicate the significance of the REE difference among sites.

The change of TF values in this study is shown in Fig. 6. The values for all REEs were below 1, indicating that *C. sinensis* had a weak ability to accumulate REEs in fruit. However, the cultivar type can largely influence REE accumulation (*Shi et al., 2022*; *Yuan et al., 2019*). *Yuan et al. (2019)* have indicated that the concentrations of REEs in grains and vegetables are significantly higher than those in fruits, including *C. sinensis*, in southern Ganzhou. Although this suggests that *C. sinensis* has a low REE accumulation capability, further studies are needed to identify the internal mechanisms of REE accumulation by *C. sinensis*. The sampling site is also vital for metal accumulation (*Yuan et al., 2019*). In this study, the TF values of LREEs were significantly higher in Wuyang compared to the
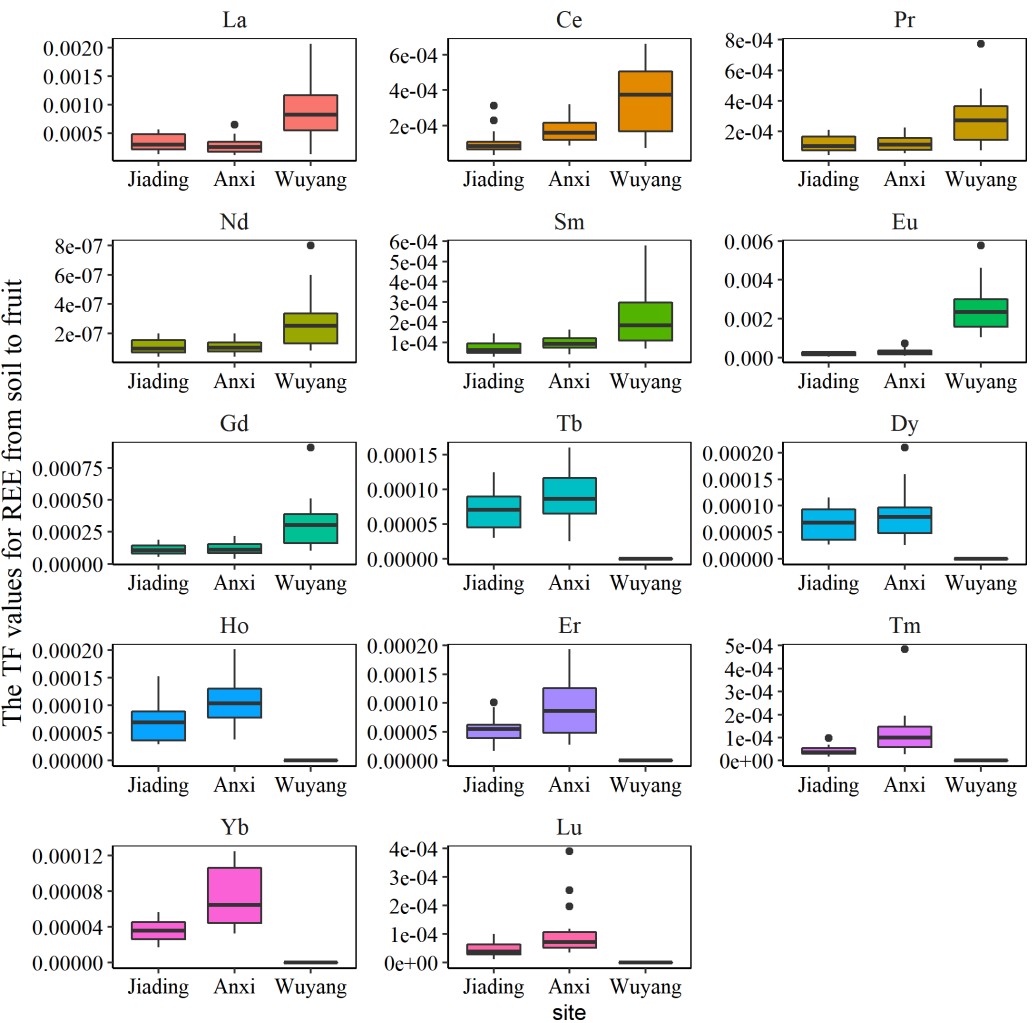

**Figure 6 The translocation factor of rare earth element from soil to the fruit of *C. sinensis*.**

other sites, whereas the TF values of HREEs were highest in Anxi, with the exception of Gd, which had higher TF values in Wuyang. As discussed above, the soil environment at different sampling sites may influence the transfer process by changing the fraction of REEs in soil, the following part will further analyze the potential mechanism.

The health risk of REE is shown in Fig. 7. The results indicated that the ADD values in Jiading, Anxi, Wuyang were all lower than 0.001 mg/kg/day, which is significantly lower than the threshold value of 0.070 mg/kg/day. This indicates that the consumption of *C. sinensis* fruits in southern Ganzhou caused no health risk to humans.

## Impacts of soil factors on rare earth elements in soils and *C. sinensis* fruit

The influence of soil properties on rare elements in soil is shown in Fig. 8. Overall, soil factors explained 28.8% of the metal difference in the studied soil, with the first two axes

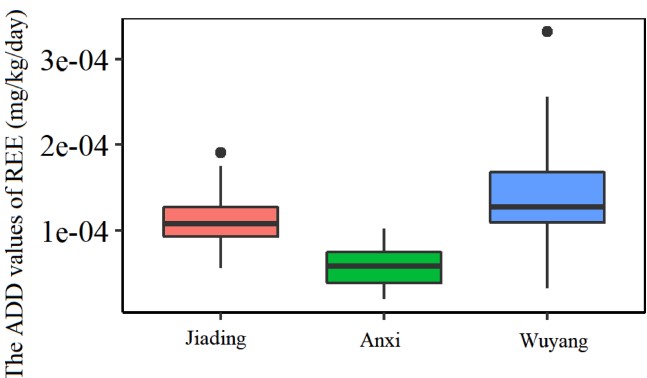

**Figure 7** **The average daily dose(ADD) of rare earth elements.**

explaining 23.75% and 0.05% of the total variance, respectively. Concretely, $K_2O$ explained 18.5% of the total variance and showed a significant positive correlation with most REE in soils ($p < 0.010$). Besides, 4.20% of the total variance of REEs in the soils was explained by TOC, which was positively correlated with the soil Ce level ($p < 0.0.50$). The influence of soil factors on the accumulation and translocation of REE in *C. sinensis* is shown in Table 5. Based on the results, $K_2O$ was positively correlated with REE in fruits and with the the TF ($p < 0.05$ or $p < 0.01$). Whereas, $Fe_2O_3$ was negatively correlated with REEs in fruits and the TF ($p < 0.05$ or $p < 0.01$). Besides, TOC negatively influenced most REEs in fruits ($p < 0.05$ or $p < 0.01$).

The soil environment can influence the metal accumulation in plants by changing metal fraction (*Hussain et al., 2021*; *Xu et al., 2021a*). Oxides, such as $K_2O$, can decrease the solubility of metals in the soil by increasing soil pH (*Li et al., 2016*), thus reducing the accumulation of metals in plants. However, $K_2O$ was positively correlated with the accumulation of REEs in the fruit of *C. sinensis* in this study. This was testified by the insignificant relationship between $K_2O$ and pH (Table S1) and the weak influence of pH on the accumulation of REEs in the fruit of *C. sinensis* (Table 5). A previous study has indicated that $K_2O$ is an important nutrients for plant growth (*Vatca et al., 2020*), and we therefore assume that it facilitates REE accumulation by promoting the growth of *C. sinensis*. Hence, the significantly higher $K_2O$ concentrations in soil from Jiading may have contributed to the higher HREE levels in fruits of *C. sinensis* compared with Anxi and Wuyang.

Fe(III) can stabilize metals in soil as Fe-oxide fraction, thus decreasing metal solubility and impeding metal accumulation in plants (*Xu et al., 2021a*). This may well explain the negative influence of $Fe_2O_3$ on REEs in fruit and on the TF. Hence, the significantly higher LREE concentrations in fruit from Wuyang may have been caused by the lower soil $Fe_2O_3$ concentration compared to the levels observed in Jiading and Anxi (Table 1). However, no significant relationship was discovered between $Fe_2O_3$ and LREE in soil (Fig. 8), most likely because the REEs levels used was the total concentration rather than the concentrations of different fractions.

Lai et al. (2023), *PeerJ*, DOI 10.7717/peerj.15470

**Table 5** The correlation between soil factors, rare earth elements in fruit and the translocation factor (TF).

| | La | Ce | Pr | Nd | Sm | Eu | Gd | Tb | Dy | Ho | Er | Tm | Yb | Lu |
|---|---|---|---|---|---|---|---|---|---|---|---|---|---|---|
| | | | | | | | REE in fruit | | | | | | | |
| $K_2O$ | 0.714** | −0.111 | 0.680** | 0.626** | 0.605** | 0.471** | 0.787** | 0.758** | 0.665** | 0.742** | 0.689** | 0.531** | 0.565** | 0.542** |
| $Fe_2O_3$ | −0.589** | −0.027 | −0.510** | −0.462* | −0.388* | −0.073 | −0.433* | −0.397* | −0.302 | −0.325 | −0.327 | −0.092 | −0.153 | −0.235 |
| CaO | −0.121 | −0.212 | −0.101 | −0.101 | −0.055 | 0.084 | −0.023 | −0.026 | 0.076 | 0.069 | −0.043 | −0.013 | −0.062 | −0.089 |
| pH | 0.102 | −0.081 | 0.088 | 0.069 | 0.107 | 0.244 | 0.107 | 0.096 | 0.170 | 0.133 | 0.054 | 0.046 | 0.068 | 0.077 |
| TOC | −0.464** | 0.009 | −0.528** | −0.526** | −0.576** | −0.067 | −0.546** | −0.529** | −0.450* | −0.495** | −0.531** | −0.366* | −0.403* | −0.259 |
| | | | | | | | TF | | | | | | | |
| $K_2O$ | 0.553** | 0.022 | 0.493** | 0.444* | 0.405* | 0.527** | 0.605** | 0.545** | 0.456* | 0.497** | 0.482** | 0.477** | 0.435* | 0.455* |
| $Fe_2O_3$ | −0.619** | −0.247 | −0.556** | −0.520** | −0.497** | −0.483** | −0.554** | −0.542** | −0.472** | −0.483** | −0.446* | −0.391* | −0.395* | −0.455* |
| CaO | −0.083 | −0.120 | −0.077 | −0.082 | −0.068 | −0.015 | −0.068 | −0.077 | −0.033 | −0.048 | −0.074 | −0.062 | −0.075 | −0.085 |
| pH | 0.143 | −0.001 | 0.123 | 0.107 | 0.112 | 0.231 | 0.133 | 0.113 | 0.119 | 0.116 | 0.085 | 0.093 | 0.106 | 0.105 |
| TOC | −0.173 | 0.057 | −0.205 | −0.188 | −0.200 | −0.048 | −0.277 | −0.233 | −0.190 | −0.210 | −0.242 | −0.229 | −0.238 | −0.163 |

**Notes.**
An asterisk (*) and double asterisks (**) denote statistical significance at $p < 0.050$ and $p < 0.001$, respectively.

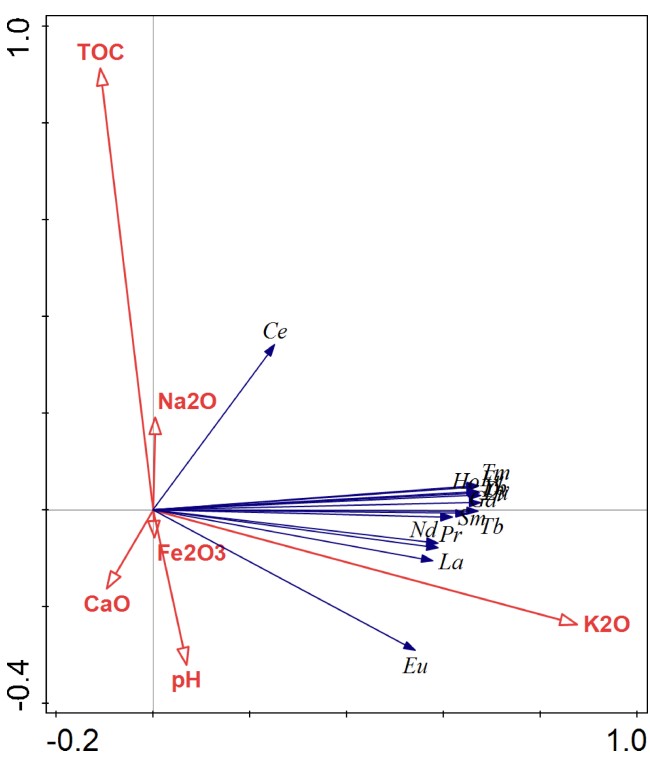

**Figure 8** The RDA analysis of the correlation between soil factors rare earth elements in soil.

Soil organic matter can often influence metal fraction in soils, thus changing metal accumulation in plants (*Zhong et al., 2020*; *Li, Huang & Mcbride, 2021*). Previous studies have indicated that soil organic matter can increase soil metal concentrations by decreasing metal availability for plants (*Luo et al., 2018*; *Piri, Sepehr & Rengel, 2019*), which is in line with the positive influence of soil organic matter on Ce in soil and may be attributed to the composition and sources of organic matter (*Wen et al., 2018*). Humic acid and EDTA can help stabilize metals in soils (*Xu, Li & Li, 2021*; *Xu et al., 2021a*; *Xu et al., 2021b*), thus decreasing metal accumulation in plants. Hence, the differences in soil organic matter composition may have caused differences in REE accumulation in fruit of *C. sinensis* in our study area. However, the composition of soil organic matter is not measured in this study, further researches are needed to identify the detailed mechanism of influence of TOC on REEs in soil and plants in the study area.

## LIMITATION AND FUTURE WORK

We analyzed the pollution, transformation, and potential risks of REEs in soil in *C. sinensis* cultivation areas, with the aim to provide information for a better management of REE-polluted soils. However, this work has some limitations that need to be addressed in future studies. First, we did not consider the different fractions of REEs, which affect the transport and transformation of REEs in the soil and their risks to humans. Second, we did not include some important environmental factors, such as soil nutrients, soil microbial

activity, and soil types, which are key components that influence the geochemical behavior and risks of REEs. Third, we did not account for the co-existence and interaction of REEs with other pollutants, such as heavy metals and organic pollutants, which may alter the behavior and risks of REEs in different media.

Based on these limitations, our future study will focus on two aspects: (1) The transport, transformation, and potential risks of REEs in soil affected by ion-adsorption deposits in southern Ganzhou, considering element fraction and key environmental factors; (2) the interaction of REEs with other pollutants and how they affect the environmental behavior and potential risks of REEs in soils.

# CONCLUSIONS

Our results indicate that the soil in *C. sinensis* orchards in southern Ganzhou is, to some extent, polluted by REEs, posing moderate ecological risks. In addition, fractionation between LREE and HREE occurred, with significant sedimentation of Ce and the loss of Eu in the studied soils. Overall, *C. sinensis* showed a weak REE accumulation capacity, posing no health risks related to the consumption of its fruit. Soil $K_2O$, $Fe_2O_3$, and TOC were key factors influencing the REE accumulation by *C. sinensis*. Our results provide a scientific basis for an improved management of the effects of mining on agricultural production.

## Funding
This paper is supported by the National Key Research and Development Program of China (2017YF0800900) and the Jiangxi Provincial Natural Science Foundation (20224BAB213036). The funders had no role in study design, data collection and analysis, decision to publish, or preparation of the manuscript.

## Grant Disclosures
The following grant information was disclosed by the authors:
National Key Research and Development Program of China: 2017YF0800900.
Jiangxi Provincial Natural Science Foundation: 20224BAB213036.

## Competing Interests
The authors declare there are no competing interests.

## Author Contributions
- Jinhu Lai conceived and designed the experiments, analyzed the data, prepared figures and/or tables, and approved the final draft.
- Jinfu Liu performed the experiments, prepared figures and/or tables, and approved the final draft.
- Daishe Wu analyzed the data, authored or reviewed drafts of the article, and approved the final draft.
- Jinying Xu conceived and designed the experiments, performed the experiments, authored or reviewed drafts of the article, and approved the final draft.

## Data Availability

The raw data is available in the Supplementary File.

## Supplemental Information

Supplemental information for this article can be found online at http://dx.doi.org/10.7717/peerj.15470#supplemental-information.

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
