# Peer review of "Pollution and health risk assessment of rare earth elements in Citrus sinensis growing soil in mining area of southern China"

_PeerJ, doi:10.7717/peerj.15470_

## Round 0.1 · original submission · Major Revisions

Dear Authors, Kindly include all suggestions given by reviewers and re-submit the revised version. Thanks.

Reviewer 1 ·

Basic reporting

This manuscript estimates the ecological and potential health risks of mining rare earth elements in Citrus sinensis growing in the vicinity of mining areas in Southern China. The work is interesting and well-organized. However, I have some observations.
1. There are several typos and grammatical errors. The work would benefit from close editing.
2. Abstract: rewrite for clarity
3. The introduction section lacks structure and doesn’t flow correctly. Consider rewriting.
4. Figure and table legends are too short. Add more descriptive captions and avoid using abbreviations.
5. Add the study’s limitations and future remarks.

Experimental design

6. Add a schematic diagram to summarize the experimental design.
7. Have the authors assessed the amount of bioavailable REEs vs the total REEs?

Validity of the findings

8. Figure and table legends are too short. Add more descriptive captions and avoid using abbreviations.
9. Consider rewriting the conclusion section.

Reviewer 2 ·

Basic reporting

The subject of the manuscript is very interesting. I think that the data contained in this paper will be a valuable addition to the literature.
However, some things need to be improved.

The language needs to be refined, both stylistically and grammatically.

In the abstract, there is no point in discussing locations named with numbers unfamiliar to the reader, but the text should be adapted to convey basic information about the research to the reader and encourage him or her to read on.

The main shortcoming of the paper is that the discussion is insufficient and too superficial and needs to be completed.

Experimental design

For the methods, it is necessary to describe the sites in more detail (why they were chosen and what their characteristics, similarities, and differences are). The sites that the authors refer to with numbers do not really make sense.

It is unclear at which sites only the soil was taken and at which the fruits of the plants were taken and why. An explanation for such behavior should be provided.

And in the text it would make more sense in some places to write which specific site it is and which characteristic (besides the property/parameter) is described.

There is a lack of information on the quality control of the results given.

Validity of the findings

The results and discussion should start with an introductory paragraph explaining what was done and why. For example, K2O is not a soil property. It is the amount of K in the soil expressed by oxides. The above points should be corrected throughout the text, especially in conclusion.

Figures are unclear what they represent.
E.g. Figure 1 represents four locations without the broader context (map, location, topography / colour, etc.).

The discussion is insufficient and too superficial and needs to be completed.
For example, the pollution status of REE evaluated by Igeo is shown in Fig. 3. In this figure, only the ranges of the determined Igeo values are listed. Nowhere are the results described in terms of Igeo pollution classes, which is the purpose of using this index.

Numbers should be reduced to three significant figures throughout the text and in all tables (e.g., 0.123; 2.34; 45.6; or 103, 1456).

The conclusion should be rephrased. It should concisely state the concluding thoughts of the study and not repeat the results. K2O is not a soil property! There is no point in listing sites, let alone naming them with numbers in the conclusion.

Additional comments

The manuscript needs to be refined in terms of language, a more detailed description of results and depth of discussion.

---

## Round 0.2 · accepted · Accept

The manuscript is accepted after minor editing.

The Secti9on Editor noted:

> There are still some language issues in the manuscript, e.g., the title and first sentence of the abstract indicate that only one rare earth element was assessed which is not correct. Also, after first-time mentioning of Citrus sinensis, in the rest of the text it should be C. sinensis. Currently, the full Latin name has been used throughout the text.

Reviewer 1 ·

Basic reporting

The authors have made substantial improvements in the revised manuscript.

Experimental design

The authors have made substantial improvements in the revised manuscript.

Validity of the findings

The authors have made substantial improvements in the revised manuscript.

Reviewer 2 ·

Basic reporting

no comment

Experimental design

no comment

Validity of the findings

no comment

Additional comments

The authors have revised the text in accordance with the suggested changes and comments, and the text has been greatly improved. As such, I believe it is suitable for publication.